# The Speed-up Factor: A Quantitative Multi-Iteration Active Learning Performance Metric

**Hannes Kath**                                                    *hannes.kath@dfki.de*
*Department of Applied Artificial Intelligence*
*University of Oldenburg*
*Oldenburg, Germany*
*Department of Interactive Machine Learning*
*German Research Center for Artificial Intelligence (DFKI)*
*Oldenburg, Germany*

**Thiago S. Gouvêa**                                              *thiago.gouvea@dfki.de*
*Department of Interactive Machine Learning*
*German Research Center for Artificial Intelligence (DFKI)*
*Oldenburg, Germany*

**Daniel Sonntag**                                                *daniel.sonntag@dfki.de*
*Department of Applied Artificial Intelligence*
*University of Oldenburg*
*Oldenburg, Germany*
*Department of Interactive Machine Learning*
*German Research Center for Artificial Intelligence (DFKI)*
*Saarbrücken, Germany*

**Reviewed on OpenReview:** *https://openreview.net/forum?id=q6hRb6fETo*

## Abstract

Machine learning models excel with abundant annotated data, but annotation is often costly and time-intensive. Active learning (AL) aims to improve the performance-to-annotation ratio by using query methods (QMs) to iteratively select the most informative samples. While AL research focuses mainly on QM development, the evaluation of this iterative process lacks appropriate performance metrics. This work reviews eight years of AL evaluation literature and formally introduces the speed-up factor, a quantitative multi-iteration QM performance metric that indicates the fraction of samples needed to match random sampling performance. Using four datasets from diverse domains and seven QMs of various types, we empirically evaluate the speed-up factor and compare it with state-of-the-art AL performance metrics. The results confirm the assumptions underlying the speed-up factor, demonstrate its accuracy in capturing the described fraction, and reveal its superior stability across iterations.

## 1 Introduction

Machine learning focuses on developing and training models that can identify general patterns from given data. These models perform exceptionally well when trained on large amounts of annotated data, earning them the term data hungry (Van Der Ploeg et al., 2014). While data collection is straightforward in many applications, data annotation tends to be time-consuming and costly. Current research aims to maximize model performance while minimizing human annotation effort.

A leading approach to reducing annotation costs is active learning (AL). Instead of randomly selecting unlabelled data samples for annotation, AL query methods (QMs) choose samples that optimize model

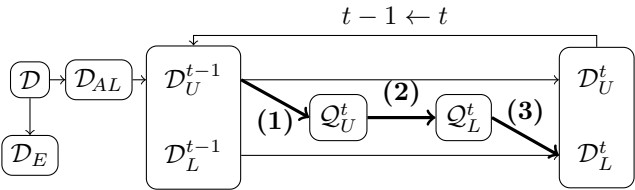

Figure 1: Active learning (AL): The dataset $\mathcal{D}$ is split into the evaluation set $\mathcal{D}_E$ and the AL set $\mathcal{D}_{AL}$. (1) A query method uses the current unlabelled dataset $\mathcal{D}_U^{t-1}$ and labelled dataset $\mathcal{D}_L^{t-1}$ to query samples $(\mathcal{Q}_U^t)$ from $\mathcal{D}_U^{t-1}$. (2) $\mathcal{Q}_U^t$ is labelled $(\mathcal{Q}_L^t)$ by a human expert. (3) Update datasets: $\mathcal{D}_U^t = \mathcal{D}_U^{t-1} \setminus \mathcal{Q}_U^t$ and $\mathcal{D}_L^t = \mathcal{D}_L^{t-1} \cup \mathcal{Q}_L^t$.

performance, improving the informativeness of the labelled dataset and enabling faster convergence with fewer annotations (Settles, 2009). Figure 1 illustrates the AL process from the data perspective. The dataset $\mathcal{D}$ is initially divided into the representative evaluation set $\mathcal{D}_E$ and the AL set $\mathcal{D}_{AL}$. At iteration $t$, $\mathcal{D}_U^{t-1}$ and $\mathcal{D}_L^{t-1}$ denote the previous unlabelled and labelled dataset. (1) A query method $\mathrm{QM}\big(\mathcal{D}_U^{t-1}, \mathcal{D}_L^{t-1}, t\big)$ selects $n\big(\mathcal{D}_U^{t-1}, \mathcal{D}_L^{t-1}, t\big)$ samples from $\mathcal{D}_U^{t-1}$ to form the unlabelled query $\mathcal{Q}_U^t$, where $n$ is the query size, also called batch size. (2) $\mathcal{Q}_U^t$ is labelled by human experts and becomes $\mathcal{Q}_L^t$. (3) Datasets are updated: $\mathcal{D}_U^t = \mathcal{D}_U^{t-1} \setminus \mathcal{Q}_U^t$ and $\mathcal{D}_L^t = \mathcal{D}_L^{t-1} \cup \mathcal{Q}_L^t$. The model $\Phi^t(\mathcal{D}_U^t, \mathcal{D}_L^t)$ is trained and the performance $P^t(\Phi^t, \mathcal{D}_E)$ on $\mathcal{D}_E$ is assessed. The next iteration begins unless a predefined stopping criterion is met. Common stopping criteria include the labelling budget $|\mathcal{D}_L^t|$, the performance level $P^t$, or the allocated annotation time.

QMs can be categorized into three types of methods: Prediction-based methods utilize model performance to select the next query, e.g., uncertainty sampling prioritizes samples near the decision boundary. Data-based methods choose samples based on the internal data structure, e.g., diversity sampling selects queries that represent the entire input space. Model-based methods focus on changes in the model, e.g., prioritizing samples with high model influence, irrespective of their labels (Tharwat & Schenck, 2023). While research on AL primarily focuses on developing more efficient QMs, relatively little attention has been paid to their empirical evaluation. QM evaluation is typically conducted by emulating AL: a labelled dataset is used, with all labels concealed from the model and revealed only when the respective samples are selected by the QM for annotation. In contrast to most machine learning tasks evaluated on final performance, AL requires evaluation across multiple iterations. Yet, the literature lacks a quantitative metric that evaluates QM performance across multiple iterations.

This work introduces the speed-up factor, a quantitative multi-iteration QM performance metric that quantifies the fraction of samples a QM needs to reach random sampling performance. Section 2 provides a literature review of the past eight years on AL evaluation. Section 3 formally introduces the proposed methods for AL evaluation. Section 4 outlines the experimental setup for our empirical evaluation. Section 5 describes the experimental results. Section 6 analyzes the results. Section 7 concludes the study.

## 2 Related Work

We provide a comprehensive overview of empirical query method (QM) evaluation through structured and unstructured literature reviews, focusing on evaluation criteria, comparison baselines, and result presentation.

The structured literature review covers proceedings from six major artificial intelligence conferences: the Association for the Advancement of Artificial Intelligence (AAAI), the Conference on Computer Vision and Pattern Recognition (CVPR), the International Conference on Learning Representations (ICLR), the International Conference on Machine Learning (ICML), the International Joint Conference on Artificial Intelligence (IJCAI), and the Conference on Neural Information Processing Systems (NeurIPS). The search criteria include publications from 2017 to 2024, with the term 'active learning' appearing in the title, abstract, or keywords.

Table 1 summarizes the quantitative findings. The selection includes 449 papers. Excluded papers are those: focusing on AL-related tasks (49); applying AL without evaluation (37); evaluating AL aspects other than

Table 1: Results of the literature review on active learning evaluation, showing the total number of papers identified through the selection criteria, those empirically evaluating a query method (QM), introducing a new QM, evaluating specific criteria, using particular baselines for comparison, and presenting evaluations in defined formats.

| | | | | Evaluation Criteria | | | Baselines | | Evaluation Presentation | | | | | |
| | | | | | | | | | On Stopping Criterion | | On Each Iteration | | Other | |
| Venue | Total | Eval QM | New QM | Perfor-mance | Process Time | Positive Samples | Random Sampling | Other QM(s) | Budget | Performance | Performance over Budget | Performance over Time | Location Samples | Ablation Study |
|---|---|---|---|---|---|---|---|---|---|---|---|---|---|---|
| AAAI | 104 | 68 | 62 | 68 | 13 | 2 | 47 | 65 | 35 | 4 | 58 | 2 | 5 | 20 |
| CVPR | 55 | 49 | 44 | 49 | 8 | 3 | 44 | 47 | 35 | 8 | 45 | 0 | 9 | 30 |
| ICLR | 48 | 35 | 33 | 35 | 9 | 0 | 32 | 33 | 21 | 2 | 31 | 0 | 9 | 14 |
| ICML | 76 | 44 | 38 | 44 | 4 | 0 | 36 | 42 | 25 | 3 | 40 | 0 | 2 | 2 |
| IJCAI | 46 | 35 | 34 | 35 | 8 | 0 | 22 | 34 | 15 | 3 | 30 | 0 | 3 | 5 |
| NeurIPS | 113 | 70 | 67 | 70 | 14 | 2 | 55 | 63 | 36 | 5 | 62 | 1 | 9 | 23 |
| Other | 7 | 5 | 2 | 5 | 1 | 0 | 5 | 5 | 2 | 1 | 5 | 0 | 0 | 0 |
| Σ | 449 | 306 | 280 | 306 | 57 | 7 | 241 | 289 | 169 | 26 | 271 | 3 | 37 | 94 |

QMs (25), e.g., model, complexity, or social factors; evaluating QMs only theoretically (22); not about AL (8), e.g., mentioning 'interactive learning'; or surveys of AL (3). A total of 306 papers empirically evaluate QMs, with 280 introducing a new QM.

## 2.1 Active Learning Evaluation Trends

**Evaluation Criteria.** All 306 papers empirically evaluating QMs include performance as an evaluation criterion, highlighting its central importance in AL. The performance of a single iteration is primarily measured using accuracy (170), F1 score (44) and mean average precision (30). Additionally, 57 papers evaluate computational requirements using QM processing time, and 7 papers use the number of positive samples selected by the QM.

**Baselines.** The performance of a QM is primarily evaluated by emulating AL and comparing it to baselines, most frequently random sampling (241).

**Evaluation Presentation.** Presenting the evaluation results of QM performance is not trivial. QMs aim to maximize the performance-to-budget ratio. The simplest approach is using either budget or performance as the stopping criterion and comparing the numerical result for the other variable against baselines in the final iteration, when the criterion is met. In our literature review, 169 papers compare the final performance using budget as the stopping criterion, while 26 papers compare the final budget using performance as the stopping criterion. While it is straightforward to present evaluations for the final iteration, there are significant caveats to this approach: The stopping criterion varies by setup, making QM results highly dependent on when AL is stopped. Performance differences between QMs change over iterations and generally decrease in later iterations as all QMs converge to the same performance level. Thus, focusing solely on results from a single iteration fails to adequately represent the overall process. A more comprehensive approach is to present the performance for each iteration. The most common method, used in 271 papers, is to present a learning curve that plots performance or loss against the size of the labelled dataset. Learning curve evaluation is performed qualitatively through visual inspection. While effective for a few clearly separated curves, visual inspection lacks quantitative, comprehensible results and poses challenges when comparing multiple QMs across different datasets (Pupo et al., 2018a). Another qualitative method, used in 37 papers, involves visualizing the locations of selected samples from a QMs in latent space, using dimensionality reduction techniques such as t-SNE or UMAP. Another quantitative method, used in 94 papers, involves evaluating specific components of a QM in isolated environments through ablation studies.

## 2.2 Active Learning Evaluation Methods

**Single-Iteration Evaluation.** Research on enhancing empirical QM evaluation primarily focuses on single-iteration assessments, presented either quantitatively for a single, typically the final, iteration or qualitatively as a learning curve. This research can be categorized into the development of new metrics, methods, or frameworks for improving empirical QM evaluation. Metric development aims to design metrics that more effectively capture QM performance than traditional measures such as accuracy or F1 score. Dayoub et al. (2017) introduce the efficiency score, defined as the ratio of test accuracy to budget size, to identify the most efficient budget size for annotation. Shi et al. (2021) introduce the label-wise average precision improvement, defined as the difference between the average precision of the QM and the baseline, divided by the average precision of the baseline. Rather than introducing a new performance metric, five papers in our literature review (e.g., (Hartford et al., 2020; Kim et al., 2021)) present the percentage performance improvements of QMs over random sampling, instead of presenting metric values as learning curves. Method development aims to enhance the validity of the evaluation process. Kong et al. (2022) argue that due to the abundance of large unlabelled datasets in AL scenarios, evaluation should be compared to semi-supervised learning, which incorporates both labelled and unlabelled data, rather than to supervised learning. Other papers use statistical significance tests, such as the t-test (Hu & Zhang, 2018; Mohamadi et al., 2022) or the Mann-Whitney U-test (Bullard et al., 2019) to evaluate the significance of performance differences between QMs. Framework development aims to provide clear guidelines for the comprehensive, quantitative, and reproducible derivation of results. By reimplementing AL experiments, Munjal et al. (2022) find that reliable empirical QM evaluation requires repeating experiments under varying training settings (e.g., model architecture, budget size), incorporating regularization techniques, and tuning hyper-parameters at each iteration. Building on this, Lüth et al. (2023) propose an evaluation framework that suggests assessing QMs across various datasets, including imbalanced ones, with different initial budgets and query sizes. The framework suggests tuning hyper-parameters on the starting budget and keeping them fixed, since grid-search fine-tuning at each iteration, as proposed by Munjal et al. (2022), is impractical due to high computational requirements. The framework additionally evaluates QMs against self-supervised learning, semi-supervised learning, and their combinations with QMs.

**Multi-Iteration Evaluation.** Riis et al. (2022) highlight the lack of a metric to assess QM performance across iterations, without offering a solution. Simple approaches for quantitative metrics that cover multiple iterations include averaging the learning curve (Tang & Huang, 2022; Gao et al., 2018) and calculating the area under the learning curve (Cawley, 2011; Pupo et al., 2018b; Pupo & Ventura, 2018). Pupo et al. (2018a) observe that these metrics neglect crucial information about intermediate results and proposes a more sophisticated evaluation method. Based on the assumption that the performance difference between two QMs ideally increases over successive iterations, this difference is calculated for each iteration (termed cut-point), and the iterations are ranked according to the magnitude of this difference. A QM is considered superior if a statistically significant correlation exists between the obtained ranking and the ideal descending ranking. While these metrics quantify results across multiple iterations, Kath et al. (2024b) highlight their high sensitivity to the number of evaluated iterations. Once the saturation point is reached, performance tends to stabilize. Subsequent iterations continuously equalize and increase both the mean value and the area under the learning curve for different QMs, and also randomize the ranking of performance differences, resulting in low stability of the metrics when evaluated across different iteration numbers. Without presenting theoretical or empirical validation, Kath et al. (2024b) use a metric similar to the speed-up factor to evaluate AL performance on bioacoustic data. Using the approximation function $\hat{p}(x) = a\left(1 - e^{-\frac{x}{b}}\right)$, where $\hat{p}$ represents the performance, $a$ the ceiling performance (computed using all samples for training), and $x$ the number of labelled training samples, the scaling parameter $b$ is fitted to the scatter points of the experimental results using least squares. After fitting the results of a QM and of the random sampling baseline, $\frac{b_{\mathrm{qm}}}{b_{\mathrm{rand}}}$ denotes the fraction of samples the QM requires to match the performance of random sampling.

## 2.3 Motivation for a new Active Learning Metric

Summarizing the trends in our literature review, the empirical evaluation of QMs primarily focuses on the performance criterion, presented either through quantitative results of a single iteration, which fail to

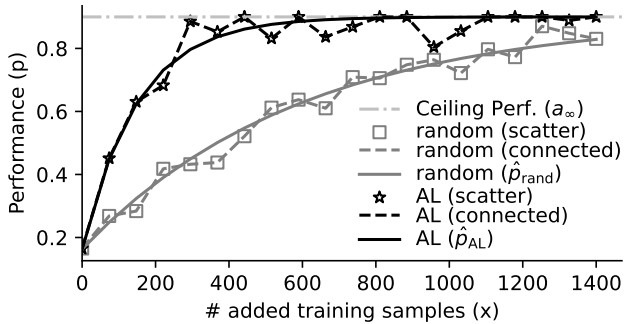

Figure 2: Learning curve schematic for random sampling and active learning (AL) showing scatter points, approximation by direct connection, and approximation by using $\hat{p}_{\mathrm{qm}}(x) = a_\infty \left(1 - e^{a_0 - \frac{x}{b_{\mathrm{qm}}}}\right)$ with $a_\infty = 0.9$, $a_0 = -0.2$, $b_{\mathrm{rand}} = 600$ and $b_{\mathrm{AL}} = 150$.

reliably represent the overall process, or through qualitative visual inspections of learning curves across iterations, which lack quantitative and comprehensible results. While metrics such as accuracy, F1 score, and mean average precision effectively capture a model's performance for a single iteration, the field of active learning lacks a comprehensive metric for quantitatively comparing results across multiple iterations. Empirical evaluation of a QM typically involves comparing its performance against baselines, predominantly using random sampling. The speed-up factor is a quantitative performance metric for evaluating QMs over multiple iterations, indicating the fraction of samples required to match random sampling performance. It generalizes the metric used by Kath et al. (2024b) to various types of learning curves by employing a set of approximation functions and provides comprehensive theoretical and empirical evidence supporting its suitability as an AL performance metric.

## 3  Methods

The performance of a machine learning model trained on a labelled dataset of a given size can be effectively represented by various performance metrics, e.g. accuracy. During the AL process, the size of the labelled dataset increases iteratively, resulting in a performance value for each iteration. QM performance is typically visualized by plotting the performance values against the labelling budget for each iteration, as shown in Figure 2. Although the data points are discrete, they are commonly visualized as a continuous graph by approximating them through direct point connection—referred to here as the learning curve (connected)—as seen in, e.g., (Dayoub et al., 2017; Gao et al., 2018; Hartford et al., 2020; Hu & Zhang, 2018; Bullard et al., 2019; Cawley, 2011). This paper introduces a novel method to approximate the learning curve, enabling the derivation of quantitative results. Our method is grounded on four assumptions:

$\mathcal{A}1$:  $t = 0 \quad \implies p_{\mathrm{qm}}(x(t)) = c_1, \ \forall \mathrm{qm} \in \mathrm{QM}, \ c_1 \in [0, 1]$

$\mathcal{A}2$:  $\mathcal{D}_U^t = \emptyset \implies p_{\mathrm{qm}}(x(t)) = c_2, \ \forall \mathrm{qm} \in \mathrm{QM}, \ c_2 \in [0, 1]$

$\mathcal{A}3$:  $p_{\mathrm{qm}}(x(t)) > p_{\mathrm{qm}}(x(t-1)), \forall \mathrm{qm} \in \mathrm{QM}, t \in \{1, 2, ..., t_{\max}\}$

$\mathcal{A}4$:  $\frac{x_{\mathrm{qm}}(p)}{x_{\mathrm{rand}}(p)} \approx c_{3,\mathrm{qm}}, \ \forall \mathrm{qm} \in \mathrm{QM}, \ \forall p \in P$

$\mathcal{A}1$ states that the performance of the initial iteration is the same for all QMs. $\mathcal{A}2$ states that the performance is the same for all QMs if the unlabelled dataset is empty. Before starting the AL process ($t = 0$) and when $\mathcal{D}_{AL}$ is completely labelled ($\mathcal{D}_U^t = \emptyset$), the composition of $\mathcal{D}_L^t$ is independent of the QM. This results in identical trained models and, consequently, the same performance.

$\mathcal{A}3$ states that the learning curve is strictly monotonically increasing. While several factors may cause deviations from this strict monotonicity, the error introduced by this assumption is minimal, as model performance generally improves with the addition of training data.

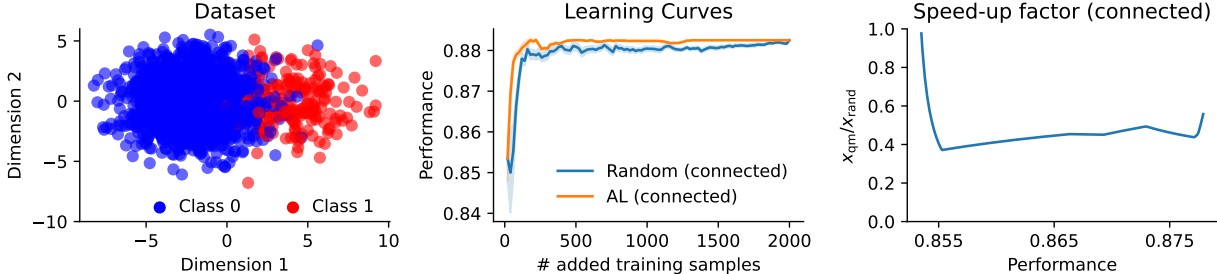

Figure 3: Synthetic illustration of $\mathcal{A}4$. Left: Two-dimensional dataset with $4\,000$ samples and 90/10 class imbalance. Center: Connected learning curves for random sampling and the query method ratio max using logistic regression. Right: Ratio $\frac{x_{\text{qm}}(p)}{x_{\text{rand}}(p)}$ obtained by inverting the connected learning curves.

$\mathcal{A}4$ states that the ratio of samples required by a QM to achieve a certain performance, compared to random sampling, is approximately constant across performance levels. Here, $c_{3,\text{qm}}$ represents this theoretical fraction, which the proposed speed-up factor $S$ estimates empirically. This assumption is based on the hypothesis that the information content of queries selected by a QM remains roughly constant. While this trend is qualitatively observable in prior work (e.g., (Kath et al., 2024c; Rakesh & Jain, 2021; Cai et al., 2021)), we are not aware of any quantitative examination of it. At most, this ratio's constancy can be inferred from tables reporting the number of samples required to reach a given performance (e.g., Table 1 in (Gal et al., 2017)). However, such data are rarely provided; as Table 1 shows, only 26 of 449 papers use performance as a stopping criterion. To further illustrate $\mathcal{A}4$, we generate a synthetic two-dimensional dataset with $4\,000$ samples and two classes in a 90/10 balance (Figure 3, left). Using a representative subset of $2\,000$ samples as the evaluation set, we emulate AL on the remaining $2\,000$ samples with an initial budget and query size of 20 using a logistic regression model. We then compute the connected learning curves for random sampling and for the QM ratio max (Figure 3, center), which is described in section 4.2. After inverting these curves, we compute the ratio $\frac{x_{\text{qm}}(p)}{x_{\text{rand}}(p)}$ (Figure 3, right). As the labelled set is independently composed before AL begins ($\mathcal{A}1$) and after all samples have been labelled ($\mathcal{A}2$), the performance at these endpoints is equal and the ratio therefore approaches 1. Figure 3 (right) confirms this behaviour at the initial point, whereas the curve does not reach 1 at the final point because variance in the learning curve (Figure 3, center) induces instability in the inversion near the endpoints. In the intermediate region, however, the ratio remains approximately constant, supporting $\mathcal{A}4$.

Instead of connecting the data points to create the learning curve, we propose to approximate it using a function $\hat{p}_{\text{qm}}(x) = \hat{p}(\frac{x}{b_{\text{qm}}})$, where $x$ is the number of training samples added to $\mathcal{D}_L^0$, $b_{\text{qm}}$ is a QM-specific parameter controlling the curve shape, $\hat{p}$ is an approximation function and $\hat{p}_{\text{qm}}(x)$ is the approximated performance value (see Figure 2). To satisfy $\mathcal{A}1$–$\mathcal{A}4$, $\hat{p}$ must be independent of $b_{\text{qm}}$ and in the range $[0, 1]$ for $x = 0$ ($\mathcal{A}1$), have a constant asymptote for $\lim_{x \to \infty}$ in range $[0, 1]$ ($\mathcal{A}2$), and be strictly monotonically increasing ($\mathcal{A}3$). Since $x$ is always divided by $b_{\text{qm}}$, $\mathcal{A}4$ is inherently satisfied. This follows directly from the definition of the speed-up factor $S$, which represents the fraction of samples required by a QM to achieve the same model performance as when using random sampling ($\hat{p}_{\text{qm}} = \hat{p}_{\text{rand}}$).

$$S = \frac{x_{\text{qm}}}{x_{\text{rand}}} = \frac{\hat{p}^{-1}(\hat{p}_{\text{qm}}) \cdot b_{\text{qm}}}{\hat{p}^{-1}(\hat{p}_{\text{rand}}) \cdot b_{\text{rand}}} = \frac{b_{\text{qm}}}{b_{\text{rand}}} = \text{const.} \tag{1}$$

Considering these restrictions, $\hat{p}$ can be freely selected based on the experimental setup. The following experiments use a set of the two most frequently used performance approximation functions (Viering & Loog, 2023):

$$\hat{p}^1 = a_\infty \left(1 - e^{a_0 - \frac{x}{b_{\text{qm}}}}\right) \quad \text{and} \quad \hat{p}^2 = a_\infty \frac{1}{1 + e^{a_0 - \left(\frac{x}{b_{\text{qm}}}\right)}}.$$

Table 2: Multi-label datasets, including name, domain, number of classes (#C), class presence statistics, and the sizes of the active learning and the evaluation set.

| Dataset | Domain | #C | Class pres. [%] min | Class pres. [%] max | Class pres. [%] mean | $|\mathcal{D}_{AL}|$ | $|\mathcal{D}_E|$ |
|---------|--------|-----|-----|-----|------|------|------|
| CARInA | Audio | 37 | 5.3 | 73.1 | 26.6 | 78 931 | 19 732 |
| MS COCO | Image | 80 | 0.2 | 54.2 | 3.6 | 94 504 | 23 783 |
| Reuters | Text | 31 | 0.3 | 22.2 | 2.2 | 37 896 | 15 675 |
| Scene | Tabular | 6 | 15.1 | 22.1 | 17.9 | 1 927 | 480 |

Given 1 experimental setup and the results of $n_{\text{QM}}$ QMs, $a_\infty$ is set to the average model performance across QMs when all samples are used for training, and $a_{0,\hat{p}^1}$ and $a_{0,\hat{p}^2}$ are derived from the results of the initial iteration ($x = 0$). Since the learning curve reflects QM performance and $\mathcal{D}_L^0$ consists of samples selected randomly or from prior annotation sessions, equal across QMs, the $x$-axis of the learning curve presents the number of added training samples with $x = |\mathcal{D}_L^t| - |\mathcal{D}_L^0|$. Using least squares, $b_{\text{qm},\hat{p}^1}$ and $b_{\text{qm},\hat{p}^2}$ are estimated for all QMs. The root mean square error (RMSE) between each QM result and the approximations $\hat{p}_{\text{qm}}^1$ and $\hat{p}_{\text{qm}}^2$ is computed and averaged across QMs. The approximation function $\hat{p}$ that minimizes the RMSE is then selected to approximate AL performance.

## 4 Experiments

The experiments have three objectives: (1) validate the four assumptions $\mathcal{A}1$–$\mathcal{A}4$ underlying the speed-up factor (see section 3); (2) identify which properties must be fixed and which may vary for valid use of the speed-up factor, based on whether $\mathcal{A}1$–$\mathcal{A}4$ are agnostic to them. We extend the framework of Lüth et al. (2023) (see section 2.2), varying dataset, AL setup, model, and training properties—detailed in the remainder of this section; and (3) evaluate the stability and robustness of the speed-up factor across diverse settings. All results are averaged over multiple random seeds.

### 4.1 Datasets

Relevant dataset properties are label type, domain, number of classes, class balance, and size of $\mathcal{D}_{AL}$. As label type we choose multi-label datasets, since they are standard in real-world scenarios and generalize multi-class datasets, which in turn extend single-class datasets. Additional results on multi-class datasets are provided in appendix A. Table 2 lists selected datasets varying in domain, number of classes, class balance, and size of $\mathcal{D}_{AL}$.

- **CARInA** (Kath et al., 2022) is an audio dataset derived from read German Wikipedia articles. We use the `Complete` subset, treat phonemes as classes, segment files into 1-second snippets, and ensure no speaker overlap between $\mathcal{D}_{AL}$ and $\mathcal{D}_E$.

- **MS COCO** (Lin et al., 2014) is an image dataset with common objects.

- **Reuters-21578** (Lewis, 1997) is a text dataset of news titles and articles. We treat topics as classes, and concatenate each title with its article.

- **Scene** (Boutell et al., 2004) is a tabular dataset of image attributes with landscapes as classes.

We use two complementary experimental settings. First, we emulate AL on the complete datasets using a stop budget of 1 400 samples, which reflects a realistic annotation budget in large-scale scenarios. Second, to analyse the behaviour of performance metrics when the unlabelled dataset is exhausted, we construct a representative 2 000-sample subset (2k) for each dataset and annotate all samples to observe the effect when $\mathcal{D}_U^t = \emptyset$. Evaluation is performed on the identical held-out evaluation set $\mathcal{D}_E$ for both settings, using the macro F1 score as the performance metric per iteration to ensure equal class importance.

## 4.2 Active Learning Setup

Relevant AL properties are initial budget $|\mathcal{D}_L^0|$, query size $|\mathcal{Q}_U^t|$, stop budget, QM type, QM performance, and QM computational requirements. We set $|\mathcal{D}_L^0|$ and $|\mathcal{Q}_U^t|$ to equal values (2, 20, or 200), ensuring all classes are represented in $|\mathcal{D}_L^0|$ when feasible, to avoid initial class selection by chance. We compare different stop budgets by treating intermediate iterations as final iteration. We choose basic, state-of-the-art, and multi-label-specific QMs, selecting 5 % of the samples randomly to ensure all samples remain eligible for selection.

- **Random sampling** as baseline selects samples arbitrarily.
- **Ratio max** (Monarch, 2021) is a basic uncertainty sampling QM that computes an uncertainty score per class. We select the maximum score per sample, following Kath et al. (2024c).
- **K-means Clustering** (Monarch, 2021) is a basic diversity sampling QM. Following Kath et al. (2024a), we reduce embeddings to five dimensions using principal component analysis for improved performance, set the number of clusters equal to the query size, and select samples closest to each cluster centroid.
- **BADGE** (Ash et al., 2020) is a state-of-the-art QM that combines uncertainty and diversity sampling. Using hypothetical labels, it leverages the gradient embedding of the model as the sampling space, selecting a query of samples likely to produce high gradients across diverse neurons.
- **BALD** (Gal et al., 2017) is a state-of-the-art QM that combines uncertainty sampling and mutual information, balancing prediction entropy and expected entropy across Monte Carlo samples to select the most informative samples.
- **CRW** (Esuli & Sebastiani, 2009) is a basic multi-label-specific QM that weights class performances to sample in a round-robin manner, selecting more samples based on the uncertainty score of classes with low performance.
- **BEAL** (Wang et al., 2024) is a state-of-the-art multi-label-specific QM that uses Bayesian deep learning to derive the model's posterior predictive distribution, and an expected confidence-based acquisition function to select uncertain samples.

## 4.3 Models

Relevant model properties are architecture, complexity, and weight initialization. Selected models include both basic and state-of-the-art variants with varying architectures and complexity:

- **wav2vec 2.0** (Baevski et al., 2020) for audio,
- **ResNet-50** (He et al., 2016) with ImageNet weights for images,
- **BERT** (Devlin et al., 2019) for text, and
- **a single fully connected layer** for tabular data.

We compare three weight initialization strategies: random initialization, self-supervised learning using rotation prediction on $\mathcal{D}_{AL}$ (Gidaris et al., 2018), and transfer learning.

## 4.4 Training

Relevant training properties are training paradigms, layer status and hyper-parameters. Selected training paradigms include supervised learning (SL), SL with data augmentation (tripling $|\mathcal{D}_L^{t-1}|$), and semi-SL with pseudo-labelling (Lee, 2013). Initializing transfer learning weights, we compare fine-tuning the last $n_l$ layers (1, 2, 5, or all). Hyper-parameters include binary cross-entropy loss with logistic activation for multi-label classification, treating each output node as an independent indicator of class presence. Training proceeds for up to 500 epochs with early stopping triggered from epoch 300 onward, using a minimum delta of 0.1 and a patience of 10 epochs, with the best weights restored.

# 5 Results

Due to the large number of relevant properties, testing all possible combinations is infeasible. Therefore, we adopt a two-step evaluation strategy. In the first step (Figure 4A–B), we fix the dataset to MS COCO$_{2k}$ and the QM to ratio max, and evaluate the effects of $|\mathcal{D}_L^0|$, $|\mathcal{Q}_U^t|$, training paradigm, weight initialization, and fine-tuning depth. Each property is varied independently, using the following default configuration: $|\mathcal{D}_L^0| = |\mathcal{Q}_U^t| = 20$, supervised learning, transfer learning weights for initialization, and fine-tuning of the last layer only. In the second step, we fix these properties to their default values and evaluate across datasets, models, QMs, and stop budgets (Figure 4C–I).

## 5.1 Effect of Budget and Training Settings

Figure 4A shows the learning curves (connected), as commonly used in the literature, by varying $|\mathcal{D}_L^0|$ and $|\mathcal{Q}_U^t|$, training paradigm, weight initialization, and fine-tuning depth. At iterations where the training set is independent of the QM—i.e., at $t = 0$ and when $\mathcal{D}_U^t = \emptyset$—performance remains consistent across $|\mathcal{D}_L^0|$ and $|\mathcal{Q}_U^t|$, but varies with changes to model or training parameters. Performance generally increases over iterations.

Figure 4B shows the fraction of samples required by a QM to match random sampling performance, derived by inverting the learning curves (connected) in Figure 4A and computing $\frac{x_{qm}(p)}{x_{rand}(p)}$, called the speed-up factor (connected). It remains stable except when performance is consistently low, such as when fine-tuning all layers. We note that, for this experimental setup, the speed-up factor exceeds 1, indicating that random sampling outperforms the QM. The metric remains fully interpretable: for example, a value of 1.1 means that the QM requires 10 % more labelled samples to achieve the same performance as random sampling. As these experiments focus on validating the speed-up factor rather than the superiority of specific QMs, such cases do not compromise the metric's validity or usefulness.

## 5.2 Effect of Datasets, Models, Query Methods, and Stop Budgets

Figure 4C shows the learning curves (connected) of all QMs on all 2k datasets. While some QMs, like BEAL on MS COCO$_{2k}$, clearly outperform others, most QMs' performance is not visually distinguishable. Performance is consistent across QMs when the training set is independent of the QM and increases monotonically.

Figure 4D shows the speed-up factor (connected) derived from Figure 4C, exhibiting low volatility across QMs and datasets. While the speed-up factor (connected) provides detailed QM performance insights, it requires both the QM and random sampling to reach all performance levels to compute $\frac{x_{qm}(p)}{x_{rand}(p)} \forall p$. As shown in Table 1, most experiments use budget as the stopping criterion. Figure 4G shows learning curves (connected) on the complete datasets with a stop budget of 1400, where $\mathcal{D}_U^t \neq \emptyset$ and not all QMs reach maximum performance—e.g., BALD on MS COCO. The speed-up factor (connected) requires annotating the entire dataset with both QM and random sampling, making it impractical; therefore, we aim to approximate its mean value.

Figure 4E shows the model performance for one QM and random sampling using $\mathcal{D}_L^t \forall t$, termed the learning curve (scatter), and the approximated learning curve $\hat{p}$ as described in section 3. The speed-up factor $S$, computed as in eq. (1), is shown in the top left of Figure 4E and as a dashed line in Figure 4D, where its close approximation to $S$ (connected) is visible.

Figure 4H shows the learning curves (scatter) and $\hat{p}$ for all complete datasets treated as single-class (with all but one class discarded), illustrating the impact of varying QM performance on metric stability, shown in Figure 4I.

Figure 4F and I show the performance metric changes when varying the stop budget for the QM and dataset used in Figure 4E and H, comparing $S$ with the mean of the learning curve (LC mean), area under the learning curve (AULC) normalized by random sampling, the cut-point method (Pupo et al., 2018a), and the fixed version of the speed-up factor (Kath et al., 2024b). LC mean shows a high volatility across experiments. AULC (normalized) is stable when QM performance resembles random sampling performance

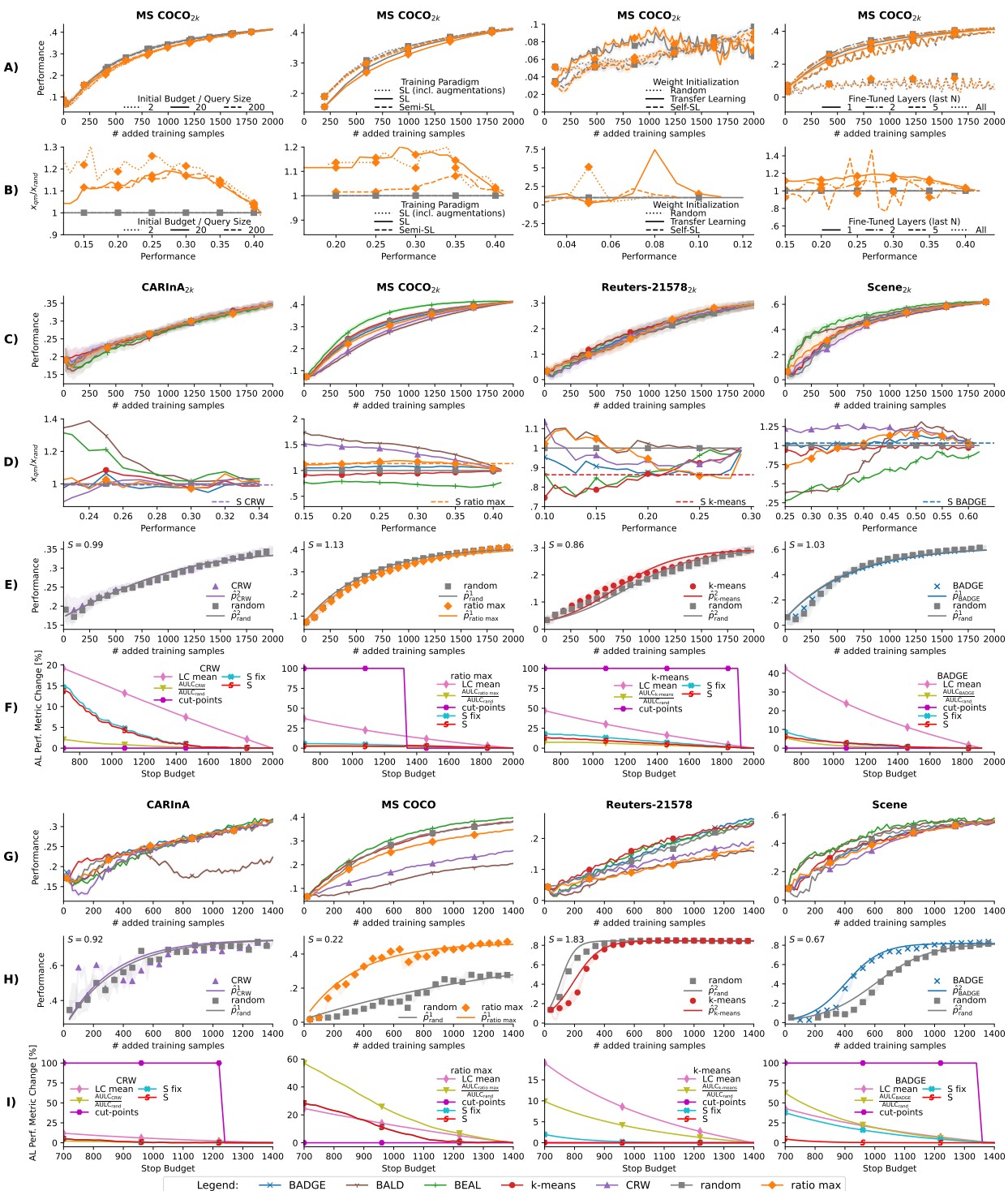

Figure 4: Empirical active learning (AL) results. 'Performance' refers to macro F1 score. Query method (QM) color and marker coding is shown in the legend below the figure; only panels with deviations include individual legends. Using multiple random seeds, $\mu$ denotes the mean, $\sigma$ the standard deviation and SEM the standart error of the mean. A–B: MS COCO$_{2k}$, all classes, QM: ratio max, 5 random seeds. **A)** Learning curve (connected), $\mu \pm$ SEM. **B)** Speed-up factor (connected), $\mu$. C–F: 2k datasets, all classes, 30 random seeds. **C)** Learning curve (connected), all QMs, $\mu \pm \sigma$. **D)** Speed-up factor (connected), all QMs, $\mu$. **E)** Learning curve (scatter + $\hat{p}$ fit), single QM, $\mu \pm \sigma$. **F)** AL Performance metric stability, single QM, $\mu$. G–I: Complete datasets, 5 random seeds. **G)** Learning curve (connected), all classes, all QMs, $\mu \pm$ SEM. **H)** Learning curve (scatter + $\hat{p}$ fit), single class, single QM, $\mu \pm$ SEM. **I)** AL Performance metric stability, single class (same as in H), single QM, $\mu$.

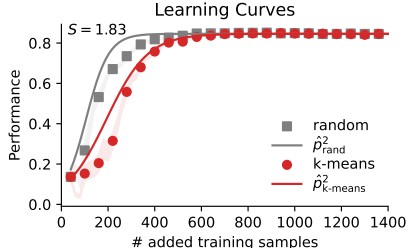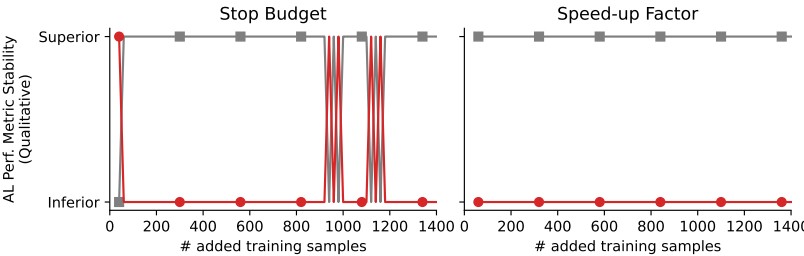

Figure 5: Metric stability comparing the stop budget method and the proposed speed-up factor. Left: Learning curves for random sampling and k-means on the single-label subset of Reuters-21578 (also shown in Figure 4H). Center: Performance decisions over the stop budget using the stop budget method. Right: Performance decisions over the stop budget using the speed-up factor.

Table 3: Assumption dependence on relevant properties. Check-marks indicate assumptions ($\mathcal{A}1$–$\mathcal{A}4$) hold despite property variation. Experiment indicate supporting figures.

| Dimension | Property | Experiment | Variation-Agnostic | | | |
|---|---|---|---|---|---|---|
| | | | $\mathcal{A}1$ | $\mathcal{A}2$ | $\mathcal{A}3$ | $\mathcal{A}4$ |
| Dataset | Label Type | Figs. 4 and 6 | ✗ | ✗ | ✓ | ✗ |
| | Domain | Fig. 4C–I | ✗ | ✗ | ✓ | ✗ |
| | # Classes | Fig. 4C–I | ✗ | ✗ | ✓ | ✗ |
| | Class Balance | Fig. 4C–I | ✗ | ✗ | ✓ | ✗ |
| | Sample Size | Fig. 4C–I | ✗ | ✗ | ✓ | ✗ |
| Active | Initial Budget | Fig. 4A–B | ✓ | ✓ | ✓ | ✓ |
| Learning | Query Size | Fig. 4A–B | ✓ | ✓ | ✓ | ✓ |
| Setup | Stop Budget | Fig. 4F&I | ✓ | ✓ | ✓ | ✓ |
| | QM Type | Fig. 4C–I | ✓ | ✓ | ✓ | ✓ |
| | QM Performance | Fig. 4C–I | ✓ | ✓ | ✓ | ✓ |
| | QM Comp. Req. | Fig. 7 | ✓ | ✓ | ✓ | ✓ |
| Model | Architecture | Fig. 4C–I | ✗ | ✗ | ✓ | ✗ |
| | Complexity | Fig. 4C–I | ✗ | ✗ | ✓ | ✗ |
| | Weight Initialisation | Fig. 4A–B | ✗ | ✗ | ✓ | ✗ |
| Training | Training Paradigm | Fig. 4A–B | ✗ | ✗ | ✓ | ✗ |
| | Layer Status | Fig. 4A–B | ✗ | ✗ | ✓ | ✗ |

(e.g., Reuters-21578$_{2k}$), but shows high volatility otherwise (e.g., MS COCO). Cut-points either accept or reject the hypothesis that the QM significantly outperforms random sampling. This is stable before the learning curve saturates (e.g., CARInA$_{2k}$), but it becomes volatile afterwards (e.g., Scene). Both $S$ (fix) and $S$ show high stability across datasets, with $S$ consistently exhibiting improved stability.

Figure 5 extends the performance stability analysis of the multi-iteration metrics from Figure 4F and I to the widely used stop budget method. Comparing the learning curves before and after saturation (Figure 5, left), the stop budget method shows instability after the learning curves saturate (Figure 5, center), whereas the speed-up factor remains stable throughout the experiment (Figure 5, right).

# 6 Discussion

We present findings on the three objectives outlined in section 4: testing assumptions $\mathcal{A}1$–$\mathcal{A}4$, identifying fixed versus variable properties, and assessing speed-up factor stability.

Results confirm the intuitive claim that performance matches at $t = 0$ and when $\mathcal{D}_U^t = \emptyset$ across QMs (Figure 4C), as the utilized training set remains unchanged—supporting $\mathcal{A}1$ and $\mathcal{A}2$. Our results consistently

show increasing learning curves, supporting $\mathcal{A}3$. Figure 4B and D show that $\frac{x_{\mathrm{qm}}(p)}{x_{\mathrm{rand}}(p)}$ remains approximately constant across all $p$, and Figure 4D and E demonstrate that its mean is well approximated by the speed-up factor, supporting $\mathcal{A}4$. The only exception occurs in Figure 4B, col. 3, where performance remains low across monitored iterations; Figure 4A, col. 4 indicates this low performance likely results from feature distortion and over-fitting caused by fine-tuning multiple pre-trained layers on small, distant datasets, whereas updating only the last one or two layers preserves general representations (Kornblith et al., 2019; Kumar et al., 2022).

Our results show that $\mathcal{A}3$ generally holds, whereas $\mathcal{A}1$, $\mathcal{A}2$, and $\mathcal{A}4$ depend on fixing dataset, model, and training properties, as summarized in Table 3. This implies that QM comparisons require identical dataset, model, and training setups, since these factors influence both the initial ($\mathcal{A}1$) and final ($\mathcal{A}2$) performance, as well as the number of samples each method needs to match random sampling performance ($\mathcal{A}4$). This dependency is expected, as performance inherently depends on the dataset, model, and training configuration, and because QMs do not perform equally well across models and tasks (Lowell et al., 2019).

Comparing metric stability under varying stop budgets (see Figure 4F and I) reveals that both the mean and area under the learning curve are sensitive to performance improvements. The cut-points metric is stable before learning curve saturation but becomes volatile afterward, as it assumes increasing performance gaps between QMs, contradicting the theoretically and empirically supported assumption $\mathcal{A}2$. The fixed and proposed speed-up factors exhibit the highest stability, with the proposed method demonstrating overall superior stability. As expected, using only the final iteration as a performance measure is unstable after the learning curve saturates (see Figure 5).

We further evaluated the sensitivity of the speed-up factor to the chosen set of approximation functions (see appendix C for details). Overall, we observe low sensitivity across datasets and QMs. Sensitivity increases for strongly non-monotonic learning curves or when performance remains consistently low until the final iteration. Based on qualitative analysis, we recommend computing the speed-up factor only when the learning curve is approximately monotonic and the normalized performance gain $\frac{P_{\mathrm{final}}-P_{\mathrm{initial}}}{P_{\mathrm{ceiling}}-P_{\mathrm{initial}}}$ exceeds approximately 20 %–30 %.

## 7 Conclusion and Limitations

### 7.1 Conclusion

This work introduces a quantitative multi-iteration AL performance metric, termed the speed-up factor. A comprehensive literature review shows that most existing AL evaluation methods are either quantitative but limited to a single iteration, or qualitative across iterations. The speed-up factor, derived from a learning curve approximation based on four assumptions, quantifies the fraction of samples a QM requires to match random sampling performance. Experiments validate these assumptions, demonstrate their agnosticism to all relevant AL setup properties, and show that the speed-up factor offers superior stability compared to existing metrics. Conclusively, the speed-up factor provides a robust, interpretable, and practically meaningful metric that enables quantitative multi-iteration evaluation of QM performance.

### 7.2 Limitations

Limitations of this work include restricting the set of approximation functions to two commonly used functions that cover key families of learning curves. Specifically, $\hat{p}^1$ corresponds to a Weibull curve with shape parameter $k = 1$, and $\hat{p}^2$ corresponds to a generalized logistic curve with shape parameter $\nu = 1$. Together, these provide a robust basis for computing the speed-up factor. While additional functions could be considered, they must satisfy the assumptions $\mathcal{A}1$–$\mathcal{A}4$, which excludes non-monotonic approximations such as unconstrained polynomials, splines, or neural network regressors.

Another limitation stems from the dependence on $\mathcal{A}1$–$\mathcal{A}4$. While $\mathcal{A}1$ generally holds, $\mathcal{A}2$ may be violated in online-learning scenarios without a fixed ceiling performance, and $\mathcal{A}3$ or $\mathcal{A}4$ can fail under unstable training, resulting in non-monotonic or highly noisy learning curves. Because the speed-up factor relies on learning curves that can be smoothly approximated and are sufficiently monotonic for reliable inversion, strong fluctuations can cause the metric to become unstable or unreliable. Smoothing or averaging across runs

can mitigate this issue, but if substantial non-monotonicity persists, we recommend not using the speed-up factor.

Furthermore, the speed-up factor becomes unreliable when performance remains consistently low. This occurs in two situations: (1) when the number of labelled samples is very small and performance remains close to the initial value, making it largely independent of the QM as stated in $\mathcal{A}1$ (see Figure 3, right); and (2) when the final-iteration performance remains substantially below the ceiling performance, causing the learning-curve approximation to be dominated by stochastic variance.

Moreover, this elaboration does not consider computational requirements; an approach to address this is detailed in appendix B.

## 8    Acknowledgments

This research is part of the Computational Sustainability & Technology project area[1], and has been supported by the Ministry for Science and Culture of Lower Saxony (MWK), the Endowed Chair of Applied Artificial Intelligence, Oldenburg University, and DFKI.

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

# A  Experiments on Multi-Class Datasets

To align with common practice in the literature, we additionally conducted experiments on multi-class datasets.

## A.1  Datasets

Table 4 lists selected multi-class datasets varying in domain, number of classes, class balance and size of $\mathcal{D}_{AL}$.

- **UrbanSound8k** (Salamon et al., 2014) is an audio dataset of urban sound categories. We segment files into 1-second snippets.

- **CIFAR-10** (Krizhevsky, 2009) is an image dataset with common objects.

- **AG News** (Zhang et al., 2015) is a text dataset of news titles and articles. We treat topics as classes, and concatenate each title with its article.

- **Letter Recognition** (Slate, 1991) is a tabular dataset of image attributes with letters (A–Z) as classes.

Table 4: Multi-class datasets, including name, domain, number of classes (#C), class presence statistics, and the sizes of the active learning and the evaluation set.

| Dataset | Domain | #C | Class pres. [%] | | | $|\mathcal{D}_{AL}|$ | $|\mathcal{D}_E|$ |
| | | | min | max | mean | | |
|---|---|---|---|---|---|---|---|
| UrbanSound8k | Audio | 10 | 4.3 | 11.5 | 10.0 | 6 971 | 1 761 |
| CIFAR-10 | Image | 10 | 10.0 | 10.0 | 10.0 | 50 000 | 10 000 |
| AG News | Text | 4 | 25.0 | 25.0 | 25.0 | 120 000 | 7 600 |
| Letter Recognition | Tabular | 26 | 3.7 | 4.1 | 3.9 | 15 857 | 4 143 |

## A.2  Experiments

Compared to the multi-label experiments, the multi-class experiments use the same active learning setup configuration. However, the model configuration differs by employing a softmax activation in the final layer instead of sigmoid, and training uses categorical cross-entropy loss in place of binary cross-entropy loss.

## A.3  Results

Figure 6 shows experimental results on multi-class datasets, applying the two-step evaluation strategy outlined in section 5.

Results and implications align with those for multi-label datasets discussed in sections 5 and 6, with the difference that multi-class classification is generally easier, as each instance belongs to a single class, simplifying decision boundaries. This leads to faster convergence of the learning curves, as shown in Figure 6A, C, E, G, and H. Even for curves with extended saturation periods, such as CIFAR-10$_{2k}$ in Figure 6C, the required sample fraction for a QM to match random sampling remains approximately constant (Figure 6D), and the proposed speed-up factor remains stable (Figure 6F).

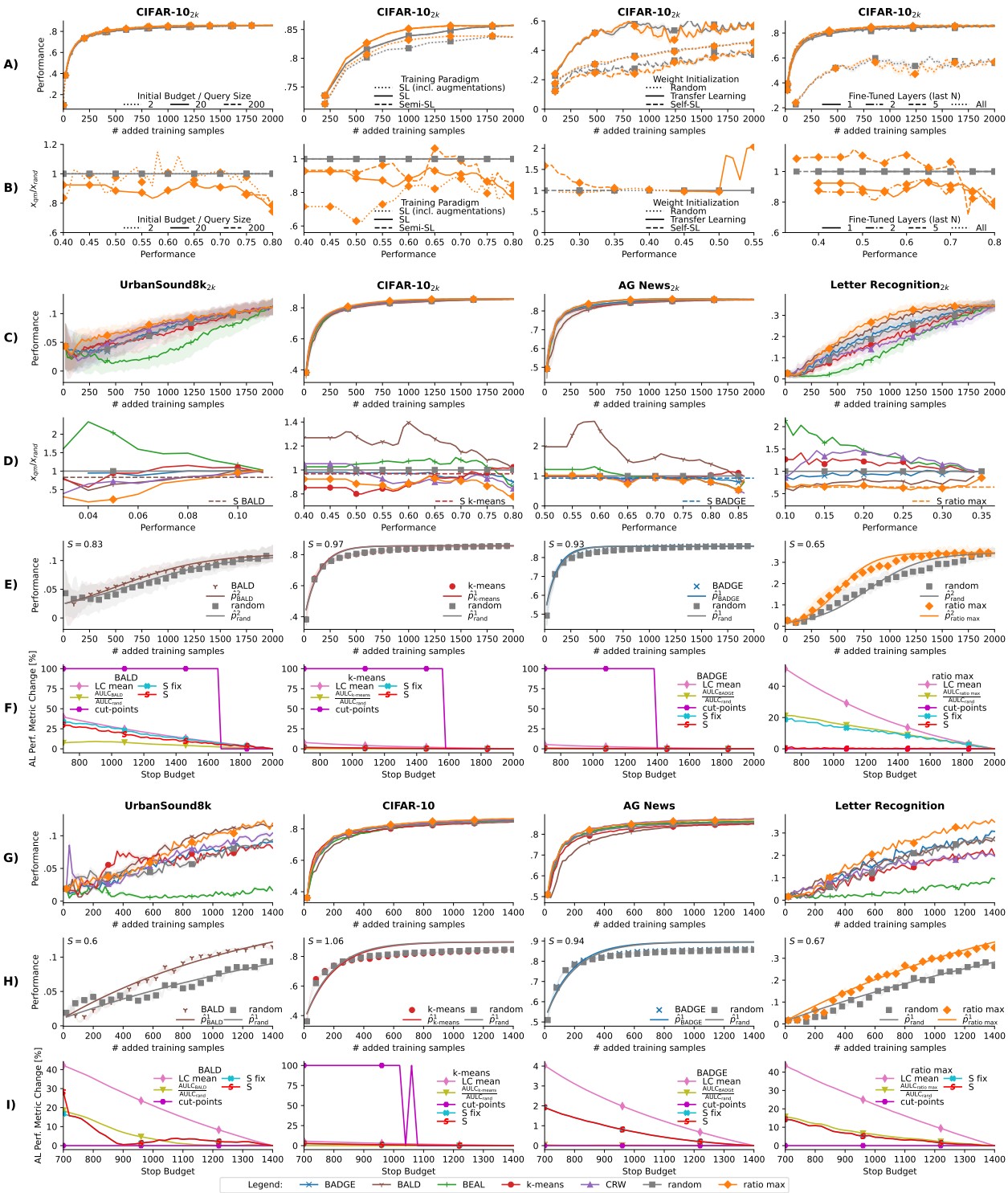

Figure 6: Empirical active learning (AL) results. 'Performance' refers to macro F1 score. Using multiple random seeds, $\mu$ denotes the mean, $\sigma$ the standard deviation and SEM the standart error of the mean. A–B: Results on CIFAR-10$_{2k}$, all classes, QM: ratio max, 5 random seeds. **A)** Learning curve (connected), $\mu \pm$ SEM. **B)** Speed-up factor (connected), $\mu$. C–F: Results on 2k datasets, all classes, 30 random seeds. **C)** Learning curve (connected), all QMs, $\mu \pm \sigma$. **D)** Speed-up factor (connected), all QMs, $\mu$. **E)** Learning curve (scatter + $\hat{p}$ fit), single QM, $\mu \pm \sigma$. **F)** AL Performance metric stability, single QM, $\mu$. G–I: Results on complete datasets, all classes, 5 random seeds. **G)** Learning curve (connected), all QMs, $\mu \pm$ SEM. **H)** Learning curve (scatter + $\hat{p}$ fit), single QM, $\mu \pm$ SEM. **I)** AL Performance metric stability, single QM, $\mu$.

## B    Evaluation of Computational Requirements

Computational requirements of a query method (QM) are critical in active learning, as they directly impact the feasibility and efficiency of iterative sample selection in real-world applications. However, Table 1 shows that these requirements are often overlooked, with only 57 of 306 papers evaluating the processing time of QMs. Since computational requirements are orthogonal to performance, they are not reflected in the speed-up factor.

### B.1    Query Methods

We propose a combined theoretical and empirical evaluation by reporting the complexity of each QM as a function of relevant factors (Table 5) and presenting normalized processing times (Figure 7).

The processing time of a QM depends on factors such as $|\mathcal{D}_U^t|$, $|\mathcal{D}_L^t|$, model complexity, number of classes, data dimensionality, and query size. We propose expressing theoretical complexity based on these dependencies to enable informed, application-specific assessments, as summarized in Table 5. Complementing the theoretical view, we propose reporting empirical processing times to capture practical efficiency. Figure 7 shows normalized processing times for the 2k subset and complete datasets used in this study.

As an example, Table 5 identifies BADGE as the most computationally expensive method, which is confirmed empirically in Figure 7. The strong dependency on $|\mathcal{D}_U^t|$ and $n_{\text{classes}}$ is evident, as processing time decreases with smaller $|\mathcal{D}_U^t|$ (Figure 4A, C; later iterations) and fewer classes (e.g., Scene$_{2\text{k}}$ and Scene).

Table 5: Complexity of query methods (QMs) in dependence of $|\mathcal{D}_U^t|$, $|\mathcal{D}_L^t|$, model complexity ($\Phi$), number of classes, data dimensions, and query size. Degree of dependence: – (none), $o(n)$ (sublinear), $O(n)$ (linear), and $\omega(n)$ (superlinear).

| QM | $|\mathcal{D}_U^t|$ | $|\mathcal{D}_L^t|$ | $\Phi$ | $n_{\text{classes}}$ | $n_{\text{dim\_data}}$ | $n_{\text{query}}$ |
|---|---|---|---|---|---|---|
| random | $o(n)$ | – | – | – | – | $o(n)$ |
| ratio max | $o(n)$ | $O(n)$ | $O(n)$ | $O(n)$ | $O(n)$ | $o(n)$ |
| k-means | $O(n)$ | – | – | – | $O(n)$ | $O(n)$ |
| BADGE | $\omega(n)$ | $O(n)$ | $\omega(n)$ | $\omega(n)$ | $O(n)$ | $O(n)$ |
| BALD | $O(n)$ | $O(n)$ | $O(n)$ | $O(n)$ | $O(n)$ | $o(n)$ |
| CRW | $o(n)$ | $O(n)$ | $O(n)$ | $O(n)$ | $O(n)$ | $o(n)$ |
| BEAL | $O(n)$ | $O(n)$ | $O(n)$ | $O(n)$ | $O(n)$ | $o(n)$ |

### B.2    Speed-Up Factor

Unlike QMs, the speed-up factor does not influence subsequent query selection and therefore only needs to be computed after the final iteration, where it is used to assess the overall performance of a QM. While it must still be evaluated by fitting the parameter $b_{\text{qm}}$ to the scattered learning curve using nonlinear least squares (e.g., `scipy.optimize.curve_fit`[2]), this computation does not introduce any per-iteration overhead within the active learning workflow.

The computational complexity of this fitting step depends on the number of parameters $\theta$ of the curve (here, $\theta = b_{\text{qm}}$ and thus $k = |\theta| = 1$), the number of data points $n$, and the number of optimization iterations $i$. Each iteration computes residuals $r_i = y_i - f(x_i, \theta)$ and processes the Jacobian matrix, resulting in a total complexity of $O(i \cdot n \cdot k + i \cdot k^3)$, which reduces to $O(i \cdot n)$ for $k = 1$.

---

[2]`https://docs.scipy.org/doc/scipy/reference/generated/scipy.optimize.curve_fit.html`

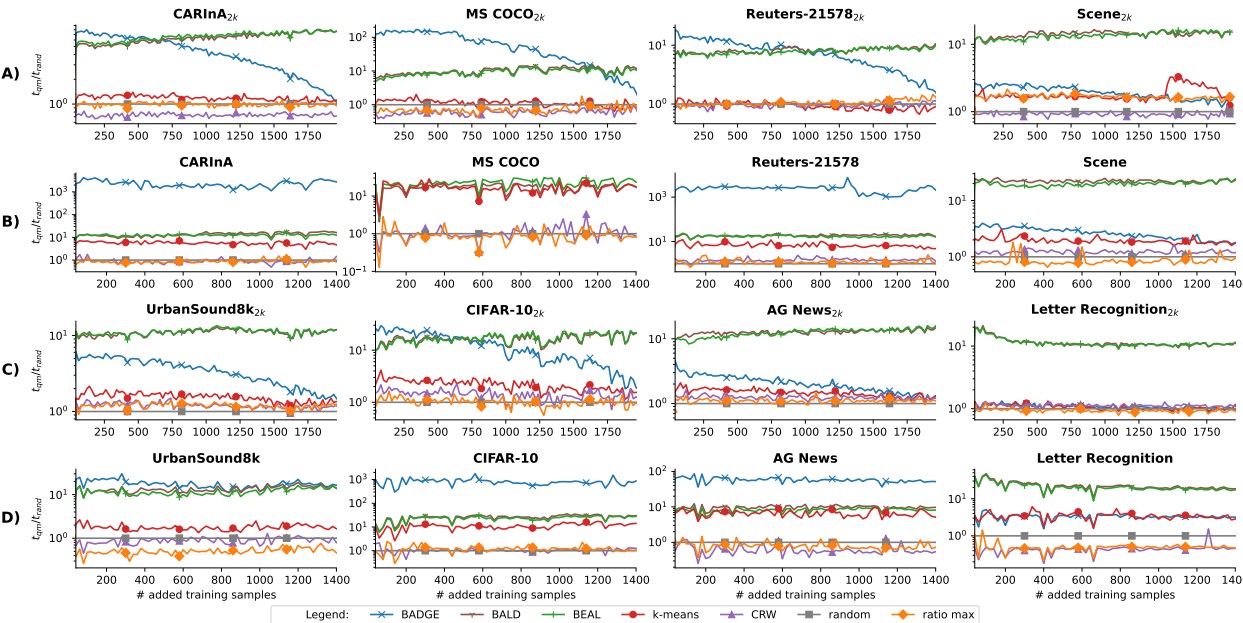

Figure 7: Normalized processing time of the active learning experiments, averaged over multiple random seeds. **A)** Multi-label 2k datasets, 30 random seeds. **B)** Multi-label complete datasets, 5 random seeds. **C)** Multi-class 2k datasets, 30 random seeds. **D)** Multi-class complete datasets, 5 random seeds.

## C   Sensitivity Analysis of the Approximation Functions

To assess the robustness of the speed-up factor $S$ with respect to the choice of approximation function, we compare the applied functions,

$$\hat{p}^1 = a_\infty \left(1 - e^{a_0 - \frac{x}{b_{\mathrm{qm}}}}\right) \quad \text{and} \quad \hat{p}^2 = a_\infty \frac{1}{1 + e^{a_0 - \left(\frac{x}{b_{\mathrm{qm}}}\right)}},$$

across all datasets and query methods. We compute two sensitivity measures: the relative difference

$$\Delta_{\mathrm{rel}} = 2 \cdot \frac{|S(\hat{p}^1) - S(\hat{p}^2)|}{S(\hat{p}^1) + S(\hat{p}^2)}$$

and the logarithmic difference

$$\Delta_{\mathrm{log}} = \big| \log(S(\hat{p}^1)) - \log(S(\hat{p}^2)) \big|.$$

Table 6 reports the quantitative results. Across all experiments on the 2k datasets, both $\Delta_{\mathrm{rel}}$ and $\Delta_{\mathrm{log}}$ remain below 5 % (with the exception of BEAL on Scene at 5.97 %), indicating that the speed-up factor is largely insensitive to the choice of approximation function when the full dataset is annotated. Experiments on the complete datasets generally show both $\Delta_{\mathrm{rel}}$ and $\Delta_{\mathrm{log}}$ below 5 %. However, comparing the sensitivity results with the learning curves shown in Figure 4G reveals that QMs with pronounced deviations from monotonic learning curves (e.g., BALD on CARInA) or low final-iteration performance (e.g., BALD on MS COCO) exhibit increased sensitivity to the choice of approximation function.

Table 6: Relative ($\Delta_{\text{rel}}$) and logarithmic ($\Delta_{\text{log}}$) differences of the speed-up factor $S$ computed using the two approximation functions $\hat{p}^1$ and $\hat{p}^2$.

| Dataset | Dataset Type | Query Method | $S(\hat{p}^1)$ | $S(\hat{p}^2)$ | $\Delta_{\text{rel}}$ [%] | $\Delta_{\text{log}}$ [%] |
|---|---|---|---|---|---|---|
| CARInA | 2k | BADGE | 0.97 | 0.98 | 0.69 | 0.69 |
| | | BALD | 1.13 | 1.10 | 2.84 | 2.84 |
| | | BEAL | 1.16 | 1.13 | 3.19 | 3.19 |
| | | k-means | 0.90 | 0.92 | 2.59 | 2.59 |
| | | CRW | 0.91 | 0.92 | 1.27 | 1.27 |
| | | ratio max | 0.95 | 0.96 | 1.28 | 1.28 |
| | complete | BADGE | 0.95 | 0.96 | 1.02 | 1.02 |
| | | BALD | 2.42 | 2.13 | 12.44 | 12.45 |
| | | BEAL | 1.15 | 1.11 | 3.95 | 3.95 |
| | | k-means | 0.88 | 0.91 | 3.56 | 3.56 |
| | | CRW | 0.76 | 0.80 | 5.12 | 5.12 |
| | | ratio max | 0.84 | 0.87 | 4.35 | 4.36 |
| MS COCO | 2k | BALD | 1.26 | 1.26 | 0.30 | 0.30 |
| | | BEAL | 0.84 | 0.87 | 3.36 | 3.36 |
| | | k-means | 0.99 | 0.99 | 0.16 | 0.16 |
| | | CRW | 1.07 | 1.06 | 0.22 | 0.22 |
| | | ratio max | 1.10 | 1.11 | 0.84 | 0.84 |
| | complete | BALD | 2.62 | 1.78 | 38.14 | 38.61 |
| | | BEAL | 0.91 | 0.93 | 2.16 | 2.16 |
| | | k-means | 1.00 | 1.00 | 0.11 | 0.11 |
| | | CRW | 1.08 | 1.06 | 2.42 | 2.42 |
| | | ratio max | 1.17 | 1.12 | 4.84 | 4.84 |
| Reuters-21578 | 2k | BADGE | 0.96 | 0.98 | 1.71 | 1.71 |
| | | BALD | 1.09 | 1.05 | 3.19 | 3.19 |
| | | BEAL | 0.93 | 0.93 | 0.04 | 0.04 |
| | | k-means | 0.95 | 0.95 | 0.03 | 0.03 |
| | | CRW | 0.95 | 0.97 | 2.77 | 2.77 |
| | | ratio max | 1.09 | 1.07 | 1.71 | 1.71 |
| | complete | BADGE | 1.00 | 1.00 | 0.11 | 0.11 |
| | | BALD | 1.80 | 1.23 | 37.66 | 38.12 |
| | | BEAL | 0.82 | 0.91 | 10.80 | 10.81 |
| | | k-means | 0.88 | 0.95 | 8.04 | 8.04 |
| | | CRW | 1.48 | 1.18 | 22.95 | 23.05 |
| | | ratio max | 1.58 | 1.21 | 26.52 | 26.68 |
| Scene | 2k | BADGE | 1.02 | 1.02 | 0.04 | 0.04 |
| | | BALD | 0.79 | 0.80 | 0.87 | 0.87 |
| | | BEAL | 0.76 | 0.81 | 5.97 | 5.97 |
| | | k-means | 0.95 | 0.97 | 1.57 | 1.57 |
| | | CRW | 1.09 | 1.09 | 0.68 | 0.68 |
| | | ratio max | 1.04 | 1.06 | 2.13 | 2.13 |
| | complete | BADGE | 1.04 | 1.03 | 0.97 | 0.97 |
| | | BALD | 0.65 | 0.72 | 10.80 | 10.81 |
| | | BEAL | 0.63 | 0.74 | 16.62 | 16.66 |
| | | k-means | 0.91 | 0.95 | 4.79 | 4.79 |
| | | CRW | 1.13 | 1.09 | 3.74 | 3.74 |
| | | ratio max | 0.98 | 1.02 | 3.40 | 3.40 |

