# OpenReview forum: "The Speed-up Factor: A Quantitative Multi-Iteration Active Learning Performance Metric"
_TMLR — Accepted by TMLR_

### Review · Reviewer_r83Y · 2025-10-20

**Summary Of Contributions:**

In their paper "The speed-up factor: a quantitative multi-iteration active learning performance metric", the authors review the literature on active learning and suggest a novel metric for quantifying the performance of an active learning algorithm ("speed-up factor"). In a nutshell, they fit exponent learning curves for the active learning algorithm and for the random sampling approach, and define speed-up factor as the ratio of the two exponents. Then they do a benchmark of active learning algorithms and argue that this new metric (speed-up factor) does make sense.

Strengths: The authors do a very thorough systematic review of existing active learning evaluations, manually annotating ~500 papers on active learning. I really liked this part. The suggested approach with the speed-up factor sounds very natural to me. The benchmark seems to be set up in a reasonable way.

Weaknesses: The suggest metric seems very obvious (disclaimer: active learning is not my field). It's also been used before (in Kath et al). The authors argue that most of the prior work uses other, more simplistic, metrics (based on thresholding) and I believe them; but they do not really show why this is important, in practice. For example, they do *not* seem to argue that their metric yields more sensible  conclusions in their benchmark, compared to more standard metrics... so the paper does not really convince me that this new metric is really necessary.

This makes me a bit on the fence about this paper. I think TMLR would be a good venue to publish a systematic review of AC evaluations, even without any new metric. But if the paper does suggest a new metric, I want more clarity on what are the benefits, especially if the authors run a whole benchmark. So I am reserving my judgment on the paper until after the authors reply.

**Audience:**

Yes

**Audience Explanation:**

Yes, the paper is on-topic for TMLR and could potentially be a good fit.

**Broader Impact Concerns:**

None.

**Claims And Evidence:**

No

**Claims Explanation:**

The claims in the review part are supported by clear evidence, yes. The claim that the new metric is a _sensible_ metric is also supported. The claim that the new metric is _better_ than the old metric is IMHO not sufficiently supported.

**Requested Changes:**

MAJOR COMMENTS

* The authors argue that the new metric (speed-up factor) is better than the stopping-criterion approaches. But they do not give any direct evidence for that. Can the benchmark results be somehow used to show that? Is the ranking of algorithms more sensible or more self-consistent using your metric compared to stopping-on-budget? Can you at least show that your metric is not fully redundant with stopping-on-budget?

* Page 5 mentions "a pool of the two" functions. This is unclear. What does a "pool" mean? How are these two functions even different? How do you choose which one to use in any given experiment? Why only these two? Two is not much of a "pool". Would always using one of them be insufficient? Can you show that?

* Section 4.1: I did not quite understand the experimental setup. Why do you do separate experiments with 1400 labeled samples and with 2000 labeled samples? Why not do everything on 2000 labeled samples? Also, why do you call the situation with 1400 labeled samples "complete"? Some datasets have |D_AL| of almost 100,000. I got very confused here, please explain this better.

* The Results section is very difficult to parse, because it's one block of text without any subsections (and the 2nd paragraph is really long), and the corresponding figure is one GIGANTIC figure with 36 panels. This is hard to follow. I would suggest to split the figure into multiple figures and the section into multiple subsections, or \paragraphs, or somehow give it more structure.

* The way you define the speed-up factor, the lower the better, right? For example, the speed-up factor in Figure is around 0.2, correct? If so, I am extremely confused by Figure 3B that shows speed-up factors above 1. Does it mean that active learning fails here? And performs worse than random sampling? When I look at Figure 3A, I don't see any big difference between active learning and random sampling, implying S of around 1. Also in panel E and in panel G the random sampling seems to perform nearly the best. I am _very_ confused by that. If active learning fails in all these cases, then why are you even showing these results? I thought the idea of active learning is that it is supposed to be more efficient than random sampling!! Please explain.

* The main conslusions of the experiments seem to be that assumptions A1-A4 are confirmed. But A1 and A2 must be true by construction, A3 must be true for any reasonable algorithm, and A4 looks like a sensible simple assumption. If all these experiments are done only to demonstrate A1-A4, then it feels underwhelming. Maybe I am missing something. See also my first bullet point.

* As the authors write, the speed-up factor is basically taken from the Kath et al paper. I don't know who the authors of this paper are, but I hope it's the same authors as in Kath et al., otherwise I feel it is a bit too bold to take a metric from Kath et al., and say that you "introduce" this metric. If it's the same or overlapping authors, then it's fine, because of course this paper does go beyond Kath et al. Unfortunately there is no way for the authors to answer this point without reavealing their identities, which is not allowed. So please do not answer this bullet point. Still, I wanted to say it.


MINOR COMMENTS

* All citations should be changed to use \citep and \citet as appropriate. Currently it looks like \citet (or maybe just \cite) was used everywhere which is not according to TMLR style guide.

* The first time "AL" appears in the Introduction it should be deabbreviated ("active learning (AL)").

* page 5: c_{3, qm} (in assumption A4) and S (in Equation 1) seem to be the same thing. Why two different notations? That's confusing.

---

> ### Author Response · Authors · 2025-11-28
> **Revisions in Response to Reviewer Feedback**
>
> Dear reviewer r83Y,
>
> Thank you for your thoughtful and balanced review. We appreciate your positive assessment of our systematic evaluation and the natural appeal of the proposed metric, as well as your constructive concerns regarding its practical benefits. Below, we address each of your points in detail.
>
> MAJOR COMMENTS
>
> 1.	We thank the reviewer for this comment. We added results to the revised manuscript showing that the stopping-criterion method becomes unstable after learning curve saturation, whereas the speed-up factor remains consistent, highlighting that our metric provides additional, non-redundant information.
>
> 2.	We thank the reviewer for pointing this out. In the revised manuscript, we now refer to a “set of functions” instead of a “pool” to avoid ambiguity. The two functions differ in their curve shapes, and for each experiment we fit both and select the one with the lower RMSE, as described in the Methods section. We chose these two functions because they are standard in the learning-curve literature (the first corresponds to a Weibull curve with shape parameter k = 1, and the second to a generalized logistic curve with shape parameter nu = 1). We added a sensitivity analysis in the appendix and discussed it in the Discussion section. The results show that the speed-up factor is generally not sensitive to the choice of approximation function, with higher sensitivity only for strongly non-monotonic learning curves or when performance remains consistently low. The revised manuscript also states explicitly in the limitations how the set of approximation functions can be extended.
>
> 3.	We thank the reviewer for highlighting the lack of clarity in the original description. The two experimental settings serve different purposes and therefore differ in size. The 1400-sample budget is used when running active learning on the complete datasets, reflecting a realistic annotation limit in large-scale scenarios rather than implying that the dataset itself is fully labelled using query methods. In contrast, the 2k setting is a deliberately constructed small (2000 samples in total), subset that enables us to analyse metric behaviour when the unlabelled dataset becomes empty. We have revised Section 4.1 to clarify this distinction.
>
> 4.	We thank the reviewer for the suggestion regarding the structure of the Results section. In the revised manuscript, we have restructured the section and introduced subsections and paragraphs to improve readability and guide the reader through the analysis. We have kept the figure as a single, comprehensive panel because we believe this allows for better comparison across experiments. However, if the reviewers feel strongly, we are happy to split the figure into multiple smaller panels.
>
> 5.	We thank the reviewer for raising this important point. As the speed-up factor quantifies the fraction of samples a query method requires to match the performance of random sampling, values greater than 1 indeed indicate that the query method performs worse than random sampling in that specific setting. Such behaviour is not unusual: prior work has shown that query methods are often sensitive to the datasets, models, and training configurations on which they are evaluated (e.g., https://papers.nips.cc/paper_files/paper/2023/hash/1ed4723f12853cbd02aecb8160f5e0c9-Abstract-Conference.html), as well as to dataset characteristics such as the number of classes (https://link.springer.com/chapter/10.1007/978-3-031-70893-0_9). Accordingly, in our experiments some configurations do not outperform random sampling for certain datasets or training setups, which naturally results in speed-up factors larger than 1. We consider these outcomes scientifically relevant, as they illustrate that the speed-up factor behaves consistently in both directions—indicating improvements when < 1 and degradations when > 1.
>
> 6.	We thank the reviewer for this comment. While A1–A4 may appear intuitive, the literature currently lacks a practical, iteration-wise evaluation metric for active learning, and it is not guaranteed that these assumptions hold across datasets, models, query methods, and training setups. Our experiments therefore serve to empirically verify that the proposed metric behaves reliably under realistic conditions and to illustrate where existing evaluation practices fail. This, in our view, aligns with the TMLR review criteria, which emphasise technical soundness and thorough empirical analysis over novelty alone.

---

> > ### Author Response · Authors · 2025-11-28
> > **Revisions in Response to Reviewer Feedback (Minor Comments)**
> >
> > MINOR COMMENTS
> >
> > 1.	We thank the reviewer for pointing this out. In the revised manuscript, all citations have been updated to use \citep and \citet as required by the TMLR style guide. We no longer use \cite anywhere in the document.
> >
> > 2.	We thank the reviewer for noting this. The revised manuscript has been updated accordingly.
> >
> > 3.	We thank the reviewer for pointing this out. c_{3,qm} in A4 represents the theoretical, assumed fraction of samples required by a query method relative to random sampling, while S is the empirically computed speed-up factor based on the fitted learning curve. To reduce confusion, we have added a clarifying sentence in the manuscript explicitly connecting these two quantities.

---

> > > ### Comment · Reviewer_r83Y · 2025-12-12
> > >
> > > Thank you for the revision and the replies. I have to stay I am still somewhat confused by this:
> > >
> > > > values greater than 1 indeed indicate that the query method performs worse than random sampling in that specific setting. Such behaviour is not unusual [...] Accordingly, in our experiments some configurations do not outperform random sampling for certain datasets or training setups, which naturally results in speed-up factors larger than 1.
> > >
> > > I understand that this _can_ sometimes happen. But I would have expected that you choose the datasets and setups where it does not happen. For example, take current Section 5.1 (Figure 4A-B). You investigate different parameters on MS COCO. But the text in Section 5.1 says _nothing_ about the fact that **all** speed-up factors are above 1 here! Could you at least add some explicit comments that? Like, I would expect to see some sentences in the end of Section 5.1 saying "Oops, in this case it turns out that all speed-up factors are above 1. That's really too bad! But whatever, in this paper we only care about how to compute the speed-up factor and not whether it is below or above 1". Please add some explicit discussion like that to 5.1. Otherwise, as a reader, I find it very confusing.
> > >
> > > Also, looking now at Figure 4, I feel lots of legends are missing. What are the grey lines in panel A? Not labeled. What are the green lines in panel C-D? Not labeled. Please make sure that ALL LINES are labeled or explained in the caption.

---

> > > > ### Author Response · Authors · 2025-12-12
> > > > **Revisions in Response to Reviewer Feedback**
> > > >
> > > > Dear reviewer r83Y,
> > > >
> > > > Thank you for the constructive remarks. Below we address your two points:
> > > >
> > > > 1.	I understand that this can sometimes happen. But I would have expected that you choose the datasets and setups where it does not happen. For example, take current Section 5.1 (Figure 4A-B). You investigate different parameters on MS COCO. But the text in Section 5.1 says nothing about the fact that all speed-up factors are above 1 here! Could you at least add some explicit comments that? Like, I would expect to see some sentences in the end of Section 5.1 saying "Oops, in this case it turns out that all speed-up factors are above 1. That's really too bad! But whatever, in this paper we only care about how to compute the speed-up factor and not whether it is below or above 1". Please add some explicit discussion like that to 5.1. Otherwise, as a reader, I find it very confusing.
> > > >
> > > > Author response: We thank the reviewer for this suggestion. We updated Section 5.1 to explicitly note that, for this experimental setup, the speed-up factors exceed 1 and therefore indicate lower performance. We clarify that the focus is on validating the speed-up factor itself, and values above 1 are fully interpretable and do not affect the metric’s validity.
> > > >
> > > > 2.	Also, looking now at Figure 4, I feel lots of legends are missing. What are the grey lines in panel A? Not labeled. What are the green lines in panel C-D? Not labeled. Please make sure that ALL LINES are labeled or explained in the caption.
> > > >
> > > > Author response: We thank the reviewer for pointing this out. The legend indicating the color and marker coding for all query methods is located below the figure, and only panels that deviate from this global coding include individual legends. We have increased the legends font size and added a clarifying sentence to the figure caption in the revised manuscript.

---

### Review · Reviewer_SEBr · 2025-10-28

**Summary Of Contributions:**

This paper introduces the speed-up factor, a new quantitative metric for evaluating the performance of active learning query methods across multiple iterations. It measures how many samples a method needs to reach the performance achieved by random sampling, providing an interpretable notion of labeling efficiency. The authors support the metric with theoretical assumptions, validate it on diverse datasets and query methods, and show that it is more stable and reproducible than existing metrics.

**Audience:**

Yes

**Audience Explanation:**

Active learning is an important area in the community. This paper proposes a method for evaluation.

**Claims And Evidence:**

Yes

**Claims Explanation:**

The paper provides extensive empirical evaluations to support the claims.

**Requested Changes:**

Questions:

1.	The authors chose two common functions to approximate the learning curve. It would be beneficial to include a brief discussion on the metric's sensitivity to this choice. For instance, what happens if neither function provides a particularly good fit (e.g., in cases of non-standard learning curve shapes)?
2.	Did the authors consider other alternative approximation functions beyond the two cited? If so, could other curve-fitting models (e.g., polynomial, spline, or neural network regressors) improve the approximation robustness compared to the chosen forms?
3.	How would the speed-up factor behave when AL is applied to non-monotonic or noisy learning curves (e.g., in unstable training scenarios)?
4.	The paper mentions that when the performance remains consistently low, the speed-up factor cannot be reliably computed. Could you provide more specific, quantitative guidance for identifying when the speed-up factor is unreliable? For instance, is there a performance threshold below which the metric should be used with caution?

Requested Changes:

1.	Several minor language and formatting issues appear in the manuscript. For example, in Section 5, “most QMs’s performance” should be corrected to “most QMs’ performance.” In addition, figure and table references are inconsistently formatted (e.g., “figure 3A” vs. “Figure 3A”, “Table” vs. “table”), and should be standardized throughout the paper.

2.	While the paper analyzes the computational complexity of query methods, it lacks an explicit discussion of the computational overhead introduced by calculating the new metric, the speed-up factor itself. Providing a complexity analysis is essential for readers to fully assess its practical implementation.

---

> ### Author Response · Authors · 2025-11-28
> **Revisions in Response to Reviewer Feedback**
>
> Dear reviewer SEBr,
>
> Thank you for your thoughtful and positive evaluation of our submission. We appreciate your recognition of the clarity and relevance of our contributions, as well as your acknowledgment of the empirical evidence supporting our claims. Below, we provide detailed responses to your specific comments.
>
> Questions:
>
> 1.	The authors chose two common functions to approximate the learning curve. It would be beneficial to include a brief discussion on the metric's sensitivity to this choice. For instance, what happens if neither function provides a particularly good fit (e.g., in cases of non-standard learning curve shapes)?
>
> Author response: thank the reviewer. We thank the reviewer for this suggestion. We added a sensitivity analysis of the metric to the appendix, demonstrating that the speed-up factor is generally robust to the choice of approximation function. We also note in the discussion that sensitivity increases for non-monotonic learning curves or when performance remains consistently low until the final iteration.
>
> 2.	Did the authors consider other alternative approximation functions beyond the two cited? If so, could other curve-fitting models (e.g., polynomial, spline, or neural network regressors) improve the approximation robustness compared to the chosen forms?
>
> Author response: We thank the reviewer for this question. While alternative curve-fitting models could be considered, they must satisfy assumptions A1–A4, which excludes non-monotonic curves such as unconstrained polynomials, splines, or neural network regressors. Adding suitable curves could enhance robustness, as the most appropriate curve for each experiment would be chosen (see section 3).
>
> 3.	How would the speed-up factor behave when AL is applied to non-monotonic or noisy learning curves (e.g., in unstable training scenarios)?
>
> Author response: We thank the reviewer for this question. The speed-up factor relies on assumptions A1–A4, and strong violations of these assumptions may render it unreliable. In particular, strongly non-monotonic or highly noisy learning curves can cause instability in the metric. Smoothing or averaging across multiple runs can mitigate this issue, but if substantial non-monotonicity persists, we recommend not using the speed-up factor. We have added the dependency on A1–A4 as an explicit limitation in the revised manuscript.
>
> 4.	The paper mentions that when the performance remains consistently low, the speed-up factor cannot be reliably computed. Could you provide more specific, quantitative guidance for identifying when the speed-up factor is unreliable? For instance, is there a performance threshold below which the metric should be used with caution?
>
> Author response: We thank the reviewer for raising this point. We identify two conditions under which the speed-up factor becomes unreliable: (i) when performance remains close to the initial value, where all query methods behave similarly, and (ii) when the final performance is still far from the ceiling, making the learning-curve approximation highly sensitive to stochastic variance. Based on qualitative analysis, we recommend using the metric only when the normalized performance gain — defined as the improvement from the initial to the final iteration relative to the gap between initial and ceiling performance — exceeds approximately 20–30%. We have added this recommendation and its justification to the discussion / limitations.
>
>
> Requested Changes:
>
> 1.	Several minor language and formatting issues appear in the manuscript. For example, in Section 5, “most QMs’s performance” should be corrected to “most QMs’ performance.” In addition, figure and table references are inconsistently formatted (e.g., “figure 3A” vs. “Figure 3A”, “Table” vs. “table”), and should be standardized throughout the paper.
>
> Author response: We thank the reviewer for pointing out these language and formatting inconsistencies. We have corrected the typo in “QMs’ performance” and standardized all figure and table references throughout the manuscript. We also reviewed the text for additional minor formatting issues and revised them accordingly.
>
> 2.	While the paper analyzes the computational complexity of query methods, it lacks an explicit discussion of the computational overhead introduced by calculating the new metric, the speed-up factor itself. Providing a complexity analysis is essential for readers to fully assess its practical implementation.
>
> Author response: We thank the reviewer for raising this point. The speed-up factor does not introduce any computational overhead during active learning because it does not influence query selection and is computed only once after the final iteration. We have added a clarification and a corresponding discussion of its computational cost in Appendix B (“Evaluation of computational requirements”) in the revised manuscript.

---

### Review · Reviewer_e43h · 2025-11-26

**Summary Of Contributions:**

This manuscript proposes a new quantitative metric, the speed-up factor, for evaluating active learning query methods across multiple iterations. The authors formalize the metric through a set of assumptions and describe a methodology for approximating learning curves using parametric functions.

**Audience:**

Yes

**Audience Explanation:**

Machine learning models benefit from large quantities of labeled data, yet annotation is often costly and time-consuming. Active learning seeks to improve annotation efficiency by using query methods to iteratively select informative samples.

**Claims And Evidence:**

Yes

**Claims Explanation:**

- The paper provides an extensive and well-organized review of evaluation practices in active learning. The quantitative summary table and the survey of methodologies used across major conferences offer strong motivation for the introduction of a new metric.
- The assumptions underlying the speed-up factor are presented clearly and supported by logical argumentation. The rationale aligns with observations reported in prior active learning research.
- The empirical evaluation is notably comprehensive.
- The proposed metric is intuitively interpretable, capturing the proportion of samples required for a method to match the performance of random sampling.

**Requested Changes:**

- Additional explanation, references to prior work, and empirical evidence are needed to substantiate Assumption 4, which posits that the ratio of samples required to reach a given performance level, relative to random sampling, remains approximately constant across performance thresholds. As this assumption is central to the metric, further theoretical discussion or a small synthetic example would considerably strengthen the manuscript.
- Table 3 suggests that Assumptions 1, 2, and 4 fail to hold under changes in the dataset, model architecture, or training paradigm. This indicates that the speed-up factor is not universally applicable and instead depends on a fixed experimental configuration. Although this limitation is noted, a more detailed discussion would be valuable, particularly for practitioners.
- The study evaluates only two functional forms for curve approximation (exponential and logistic). While these are common choices, the learning-curve literature encompasses broader families such as power-law or Weibull models. Including at least a sensitivity analysis, perhaps in the appendix, would improve robustness.
- Although the speed-up factor does not explicitly account for computational overhead, repeatedly fitting parametric functions may be costly in large-scale experiments. A brief analysis of the computational complexity would be helpful.

---

> ### Author Response · Authors · 2025-11-28
> **Revisions in Response to Reviewer Feedback**
>
> Dear reviewer e43h,
>
> Thank you for your thoughtful and positive assessment of our work. We appreciate your recognition of the clarity, motivation, and empirical depth of the manuscript. Below, we respond to your specific comments in detail.
>
> 1.	Additional explanation, references to prior work, and empirical evidence are needed to substantiate Assumption 4, which posits that the ratio of samples required to reach a given performance level, relative to random sampling, remains approximately constant across performance thresholds. As this assumption is central to the metric, further theoretical discussion or a small synthetic example would considerably strengthen the manuscript.
>
> Author response: We thank the reviewer for this suggestion. While we did not find a quantitative discussion of this in the literature, the trend can be qualitatively observed and quantitatively computed if budgets required to reach given performance levels are reported. We have included this discussion for A4 in Section 3 and additionally conducted a synthetic dataset experiment supporting A4, which is also described in Section 3.
>
> 2.	Table 3 suggests that Assumptions 1, 2, and 4 fail to hold under changes in the dataset, model architecture, or training paradigm. This indicates that the speed-up factor is not universally applicable and instead depends on a fixed experimental configuration. Although this limitation is noted, a more detailed discussion would be valuable, particularly for practitioners.
>
> Author response: We thank the reviewer for this helpful observation. As Table 3 indicates, Assumptions 1, 2, and 4 do not hold under changes to the dataset, model architecture, or training setup. We have expanded the discussion to provide clearer guidance for practitioners on when the speed-up factor can be applied reliably and which changes in experimental conditions require re-estimating the underlying assumptions. Under such changes, the speed-up factor is no longer comparable across configurations.
>
> 3.	The study evaluates only two functional forms for curve approximation (exponential and logistic). While these are common choices, the learning-curve literature encompasses broader families such as power-law or Weibull models. Including at least a sensitivity analysis, perhaps in the appendix, would improve robustness.
>
> Author response: We thank the reviewer for this important point. We added a sensitivity analysis of the metric to the appendix, showing that the speed-up factor is generally robust to the choice of approximation function. We also note in the discussion that sensitivity increases for non-monotonic learning curves or when performance remains low until the final iteration. Additionally, the limitations now explicitly describe how the set of approximation functions can be extended.
>
> 4.	Although the speed-up factor does not explicitly account for computational overhead, repeatedly fitting parametric functions may be costly in large-scale experiments. A brief analysis of the computational complexity would be helpful.
>
> Author response: We thank the reviewer for raising this point. We have added a clarification and a corresponding discussion of its computational cost in Appendix B (“Evaluation of computational requirements”) in the revised manuscript.

---

### Decision · Action_Editor_WSFe · 2026-01-20

**Recommendation:** Accept as is

**Audience:**

Yes

**Audience Explanation:**

The section of the community working on sample efficiency and active learning

**Claims And Evidence:**

Yes

**Claims Explanation:**

This work reviews eight years of AL evaluation literature and formally introduces the speed-up factor, a quantitative multi-iteration QM performance metric that indicates the fraction of samples needed to match random sampling performance. All the reviewers agree that the claims in the paper are supported by sufficient evidence; I too agree and hence argue for acceptance